# Mitochondrial genome of eight Carangidae and phylogenetic analysis in the family

Fangcao Zhao[1☯], Lin Xian[1,2,3☯], Kecheng Zhu[1,2,3], Nan Zhang[1,2,3], Huayang Guo[1,2,3], Baosuo Liu[1,2,3], Jingwen Yang[1,2,3], Bo Liu[1], Dianchang Zhang[1,2,3]*

1 Key Laboratory of South China Sea Fishery Resources Exploitation and Utilization, Ministry of Agriculture and Rural Affairs, South China Sea Fisheries Research Institute, Chinese Academy of Fishery Sciences, Guangzhou, Guangdong Province, People's Republic of China, 2 Sanya Tropical Fisheries Research Institute, Sanya, China, 3 Guangdong Provincial Engineer Technology Research Center of Marine Biological Seed Industry, Guangzhou, Guangdong Province, People's Republic of China

☯ These authors contributed equally to this work.

* zhangdch@scsfri.ac.cn

## Abstract

The Carangidae family is a prime focus for both deep-sea fishing and aquaculture. However, taxonomic controversies have limited Carangidae research. This study assembled the mitochondrial genomes of eight Carangidae species using second-generation sequencing and bioinformatics, then performed phylogenetic analyses. Mitochondrial genome sizes were: *Megalaspis cordyla* (16,565 bp; OR703829), *Elagatis bipinnulata* (16,543 bp; OR668919), *Scomberoides tol* (16,689 bp; OR668917), *Selaroides leptolepis* (16,560 bp; OR703831), *Decapterus maruadsi* (16,540 bp; OP459436), *Alepes kleinii* (16,570 bp; OR668918), *Caranx sexfasciatus* (16,595 bp; OR703830), and *Carangoides orthogrammus* (16,604 bp; OR668920). This study provides the first complete mitochondrial genome sequences of the species for *Scomberoides tol*, *Carangoides orthogrammus*, and *Caranx sexfasciatus*. The genomes contained two rRNA genes, 13 protein-coding genes, and 22–23 tRNAs, all with A+T bias. Phylogenetic analysis revealed a genetic distance of 0.002 between *Uraspis secunda* and *U. helvola*, suggesting that they are synonymous. The genetic distance between *A. kleinii* and *A. djedaba* was 0.082, reflecting their presence in the same genus. Intrageneric distance was greater than intergeneric distance between *C. equula* and *C. orthogrammu*, inconsistent with their taxonomic status. Finally, Seriolina and Caranginae were closely related, as were Trachinotinae and Chorineminae. In conclusion, our results provide breeding resources and an empirical basis for resolving Carangidae taxonomy.

## 1. Introduction

Mitochondrial DNA (mtDNA) possesses multiple characteristics that make it well-suited for species identification, classification, phylogeny reconstruction, germplasm

**Data availability statement:** The data that support the findings of this study are available in NCBI at https://www.ncbi.nlm.nih.gov/nuccore/OP459436, reference number OP459436.1.

**Funding:** Central Public-Interest Scientific Institution Basal Research Fund, CAFS (NO. 2022TS07).

**Competing interests:** The authors have declared that no competing interests exist.

resource analysis, and evolutionary research. These advantages include low molecular weight, strong coding ability, high copy number, rapid evolution, polymorphism, and matrilineal inheritance [1–3]. In fish, mtDNA consists of a non-coding region and 37 genes encoding 22 tRNAs, two rRNAs, and 13 proteins (protein-coding genes, PCGs) [4,5]. The gene sequences are fairly short, leading to a stable structure, and in most ostracod fishes, the gene positions are relatively fixed [6]. However, gene rearrangements are present in the mitochondrial genomes of several species, including *Cynoglossus robustus*, *Amphilophus*, and *Odontobutis platycephala* [7–9]. Therefore, certain genes may have been duplicated or mutated during fish evolution. Because of their matrilineal inheritance, mtDNA are useful for revealing phylogenetic relationships between taxonomic orders.

The Carangidae family are a group of economically important fishes that can be divided into four subfamilies: Caranginae, Seriolinae, Trachinotinae, and Chorineminae [10]. They have an extended and laterally compressed body plan but otherwise are varied in shape; their meat is valued for tenderness and high nutrition [11,12]. The extended distribution and abundance of Carangidae has led to the family becoming prominent in fisheries worldwide [13]. However, overfishing and pollution are lowering Carangidae resources, necessitating conservation and improved management [14].

Several studies have been conducted on Carangidae members. For example, mtDNA Cytb sequences have allowed for in-depth investigation of genetic variation among *Decapterus maruadsi* populations in the South China Sea [15]. Another study has successfully applied mtDNA to clarify *Decapterus russelli* phylogeny [16]. Using complete mtDNA sequences, the systematic evolution of *Pseudocaranx dentex* [17] and the reproductive status of *Seriola dumerili* in relation to other species [18] were examined, and these results mainly focuses on family. Mitochondrial 16SrRNA sequences clarified the phylogenetic relationships of 12 Carangidae species [19], and benefited research to classify Carangidae DNA barcodes, providing a reference for species identification and systematic relationship construction [20].

Despite the available research, the relationships between the four Carangidae subfamilies have long been controversial, given high frequencies of synonymy and heteronymy. Indeed, few systematic analyses have been performed on the molecular classification and phylogeny of Carangidae.

This study performed second-generation sequencing and bioinformatics analysis on the whole mitochondrial genomes of eight Carangidae species (*Megalaspis cordyla, Elagatis bipinnulatus, Scomberoides tol, Selaroides leptolepis, D. maruadsi, Alepes kalla, Caranx sexfasciatus*, and *Carangoides orthogrammus*). We then combined these sequences with mtDNA data from NCBI to yield 33 Carangidae species mtDNA genomes. We investigated genome structure, codon usage, and gene arrangement order. Additionally, we constructed three phylogenetic trees to explore Carangidae relationships and species divergence times. Our findings should provide a theoretical basis for Carangidae taxonomy, evolutionary genetics, and the development of improved germplasm resources.

## 2. Materials and methods

### 2.1. Sample and DNA extraction

In December 2020, this study collected samples of *D. maruadsi* from the waters surrounding Dongshan Dao, Fujian; *A. kleinii* and *S. leptolepis* in Beihai, Guangxi; as well as *M. cordyla*, *S. tol*, *E. bipinnulatus*, *C. sexfasciatus*, and *C. orthogrammu*. All experiments carried out in this research complied with the regulations and guidelines established by the Animal Care and Use Committee of the South China Sea Fisheries Research Institute of the Chinese Academy of Fishery Sciences and approved by (No. SCSFRI96-253). There are no ethical issues in this study. All samples were collected while alive and narcotized by MS-222. A third of the anal fins and the second dorsal fins were cut and immediately stored in ethanol (75%), then replaced twice with ethanol and stored at −20°C. All the fish samples were released after the fins were taken. Genomic DNA was obtained from the samples using a DNA extraction kit (Mobio, Guangzhou, China). Their quality was assessed via agarose gel electrophoresis and nucleic acid/protein assays. High-quality DNA was submitted for sequencing.

### 2.2 Sequencing, assembly, and annotation

Whole genome sequencing was performed by Shenzhen Huada Gene Technology. NOVOPlasty 2.6.3 with default parameters was used to extract and assemble mitochondrial genome sequences from raw genome sequence data [21]. The complete mitochondrial genome sequence was spliced and uploaded to the MITOS web server (http://mitos.bioinf.unil-eipzig.de/). Furthermore, coding genes, RNAs, and noncoding regions were annotated, codons were selected from the vertebrate database, and other parameters were set to default [22].

### 2.3 Bioinformatics analysis

Mitochondrial DNA was proofread in DNAStar and spliced to obtain the full sequence; total length, base percentage, and GC content were determined [23]. Additionally, PCGs, rRNA genes, and D-loop regions in whole mtDNA sequences were analyzed using BLAST and DNAStar. Long tandem repeats within the control region were initially identified with Tandem Repeats Finder, then reanalyzed manually [24]. Codon preference was analyzed using codon W, and the mitochondrial genome was mapped in OGDRAW 1.2 [25]. The composition of PCG clusters and the genetic distances between them were calculated in MEGA X, along with identifying genes exhibiting AT-skew and GC-skew [26].

### 2.4 Phylogenetic and molecular clock analyses

Phylogenetic analyses employed the eight mtDNA whole genomic sequences of Carangidae species obtained here, along with 34 mtDNA sequences from NCBI GenBank (http://www.ncbi.nlm.nih.gov/) (34 Carangidae species and *Lates calcarifer* as an outgroup; Table 1). Downloaded genome sequences were aligned and manually calibrated in ClustalX [27]. Sequence length and base content were determined in MEGA X. The Kimura two-parameter method was used to calculate interspecific genetic distance [28]. Protein-coding genes were individually selected for coding region sequence comparisons in mafft version 7.490 using default parameters [29]. IQ-TREE version 2.0 (http://www.iqtree.org/) [30] was used to construct a maximum likelihood (ML) phylogenetic tree with the selected sequences. The optimal model (TVM+F+R5) was selected based on Bayesian information criterion (BIC) scores; the tree's parameters were set as: -m MFP -B 1000 -alrt 1000. Next, nucleic acid modelling of selected DNA sequences was performed in jModelTest 2.1.7 [31], and the best model for tree construction was selected based on minimum AIC (Akaike Information Criterion). A Bayesian (BI) phylogenetic tree was constructed in MrBayes version 3.2.7a [32] using the optimal model GTR+I+G. We ran the BI analysis for 10,000 generations with 1000 bootstraps. A neighbor-joining (NJ) tree was then constructed [33] in MEGA 7 [34], using the p-distance method and bootstrap test (1000 replicates) [35].

**Table 1. Base composition and GenBank IDs of mitochondrial whole genomes.**

| Subfamily | Genus | Latin name | Gene Bank ID | Base composition/% | | | | | | Size/ bp |
|---|---|---|---|---|---|---|---|---|---|---|
| | | | | T | C | A | G | A+T | G+C | |
| Caranginae | Decapterus | Decapterus macrosoma | NC_023458. 1 | 25. 4 | 30. 4 | 27. 0 | 17. 2 | 52. 4 | 47. 6 | 16536 |
| | | Decapterus macarellus | NC_026718. 1 | 25. 3 | 30. 4 | 27. 3 | 17. 0 | 52. 6 | 47. 4 | 16544 |
| | | Decapterus russelli | MN711693. 1 | 25. 4 | 30. 2 | 27. 5 | 16. 9 | 52. 9 | 47. 1 | 16542 |
| | | Decapterus maruadsi | OP459436 | 25. 2 | 30. 5 | 27. 4 | 16. 9 | 52. 6 | 47. 4 | 16540 |
| | | Decapterus tabl | NC_044650. 1 | 25. 0 | 30. 6 | 27. 3 | 17. 1 | 52. 3 | 47. 7 | 16545 |
| | Trachurus | Trachurus japonicus | NC_002813. 1 | 25. 8 | 29. 9 | 27. 7 | 16. 6 | 53. 5 | 46. 5 | 16559 |
| | | Trachurus trachurus | NC_006818. 1 | 25. 8 | 29. 9 | 27. 7 | 16. 6 | 53. 5 | 46. 5 | 16559 |
| | Pseudocaranx | Pseudocaranx dentex | MZ398237. 1 | 25. 4 | 30. 2 | 27. 2 | 17. 2 | 52. 6 | 47. 4 | 16570 |
| | Selar | Selar crumenophthalmus | NC_023954. 1 | 26. 6 | 29. 5 | 27. 2 | 16. 8 | 53. 8 | 46. 3 | 16610 |
| | Caranx | Caranx sexfasciatus | OR703830 | 26. 3 | 28. 9 | 28. 9 | 15. 9 | 55. 2 | 44. 8 | 16595 |
| | | Caranx tille | NC_029421. 1 | 26. 3 | 28. 9 | 28. 9 | 15. 8 | 55. 2 | 44. 7 | 16593 |
| | | Caranx melampygus | KF649843. 1 | 26. 2 | 28. 9 | 28. 8 | 16. 1 | 55. 0 | 45. 0 | 16597 |
| | | Caranx ignobilis | NC_022932. 1 | 25. 8 | 29. 3 | 28. 8 | 16. 0 | 54. 6 | 45. 3 | 16588 |
| | Megalaspis | Megalaspis cordyla | OR703829 | 25. 8 | 29. 4 | 28. 8 | 15. 9 | 54. 6 | 45. 3 | 16565 |
| | Alepes | Alepes kleinii | OR668918 | 27. 0 | 28. 5 | 28. 1 | 16. 4 | 55. 1 | 44. 9 | 16570 |
| | | Alepes djedaba | NC_037049. 1 | 26. 4 | 29. 0 | 27. 9 | 16. 7 | 54. 3 | 45. 7 | 16563 |
| | Atule | Atule mate | NC_026222. 1 | 27. 6 | 27. 7 | 28. 4 | 16. 3 | 56. 0 | 44. 0 | 16565 |
| | Gnathanodon | Gnathanodon speciosus | NC_054367. 1 | 26. 3 | 28. 7 | 29. 4 | 15. 5 | 55. 7 | 44. 2 | 16555 |
| | Selaroides | Selaroides leptolepis | OR703831 | 26. 6 | 28. 9 | 27. 8 | 16. 8 | 54. 4 | 45. 7 | 16560 |
| | Parastromateus | Parastromateus niger | KJ192332. 1 | 26. 0 | 29. 5 | 28. 3 | 16. 2 | 54. 3 | 45. 7 | 16561 |
| | Uraspis | Uraspis secunda | NC_029488. 1 | 25. 8 | 29. 8 | 28. 2 | 16. 2 | 54. 0 | 46. 0 | 16554 |
| | | Uraspis helvola | NC_033402. 1 | 25. 8 | 29. 8 | 28. 1 | 16. 2 | 53. 9 | 46. 0 | 16555 |
| | Carangoides | Carangoides equula | NC_025644. 1 | 25. 3 | 30. 2 | 26. 3 | 18. 1 | 51. 6 | 48. 3 | 16588 |
| | | Carangoides orthogrammus | OR668920 | 25. 5 | 30. 0 | 27. 9 | 16. 6 | 53. 4 | 46. 6 | 16604 |
| | | Carangoides bajad | LC557137. 1 | 26. 1 | 29. 8 | 28. 4 | 15. 8 | 54. 5 | 55. 6 | 16556 |
| | | Carangoides plagiotaenia | NC_051884. 1 | 26. 7 | 29. 1 | 28. 2 | 15. 9 | 54. 9 | 45. 0 | 16551 |
| | | Carangoides armatus | NC_004405. 1 | 26. 5 | 29. 4 | 28. 0 | 16. 1 | 54. 5 | 45. 5 | 16556 |
| | | Carangoides malabaricus | NC_023968. 1 | 26. 2 | 29. 6 | 27. 8 | 16. 4 | 54. 0 | 46. 0 | 16561 |
| | Alectis | Alectis ciliaris | NC_025566. 1 | 26. 8 | 28. 8 | 28. 3 | 16. 2 | 55. 1 | 45. 0 | 16570 |
| | | Alectis indica | NC_037050. 1 | 25. 6 | 30. 2 | 28. 0 | 16. 2 | 53. 6 | 46. 4 | 16553 |
| Seriolinae | Seriola | Seriola rivoliana | KP733847. 1 | 25. 7 | 29. 8 | 27. 3 | 17. 2 | 53. 0 | 47. 0 | 16599 |
| | | Seriola dumerili | MZ398238. 1 | 25. 5 | 30. 0 | 26. 8 | 17. 6 | 52. 3 | 47. 6 | 16530 |
| | | Seriola lalandi | NC_016869. 1 | 25. 3 | 30. 2 | 26. 7 | 17. 8 | 52. 0 | 48. 0 | 16532 |
| | | Seriola quinqueradiata | NC_016868. 1 | 25. 2 | 30. 2 | 26. 6 | 18. 0 | 51. 8 | 48. 2 | 16537 |
| | Seriolina | Seriolina nigrofasciata | NC_028420. 1 | 25. 8 | 30. 0 | 26. 7 | 17. 5 | 52. 5 | 47. 5 | 16531 |
| | Elagatis | Elagatis bipinnulatus | OR668919 | 25. 8 | 29. 5 | 27. 9 | 16. 8 | 53. 7 | 46. 3 | 16543 |
| Trachinotinae | Trachinotus | Trachinotus blochii | NC_024026. 1 | 26. 5 | 28. 6 | 29. 2 | 15. 7 | 55. 7 | 44. 3 | 16558 |
| | | Trachinotus ovatus | KJ642220. 1 | 26. 2 | 28. 9 | 29. 0 | 15. 9 | 55. 2 | 44. 8 | 16564 |
| | | Trachinotus carolinus | NC_024184. 1 | 26. 0 | 29. 1 | 28. 7 | 16. 3 | 54. 7 | 45. 4 | 16544 |
| Chorineminae | Sco Mberoides | Scomberoides tol | OR668917 | 25. 1 | 31. 0 | 28. 3 | 15. 6 | 53. 4 | 46. 6 | 16689 |
| | | Scomberoides lysan | NC_063497. 1 | 25. 5 | 30. 5 | 28. 3 | 15. 7 | 53. 8 | 46. 2 | 16767 |
| Latidae | Lates | Lates calcarifer | NC_007439. 1 | 25. 3 | 30. 0 | 28. 6 | 16. 1 | 53. 9 | 46. 1 | 16535 |

Aligned sequences were imported into BEAST2 version 2.7.1 for molecular-clock phylogenetic tree construction [36]. Parameters were set as Subst Model: HKY, reference differentiation time: *Alectis ciliaris-Alectis indica*: 22.74 Mya (differentiation time from Timetree, http://www.timetree.org/), and number of iterations: 10,000,000.

## 3. Results

### 3.1 Genome structure and codon usage

The mtDNAs of *M. cordyla, E. bipinnulatus, S. tol, S. leptolepis, D. maruadsi, A. kleinii, C. sexfasciatus*, and *C. orthogrammu* all exhibited a typical double-stranded, closed-ring structure (Fig 1). The genome contained 37–38 genes ranging from 16540 bp to 16689 bp in length, including 13 PCGs, 22–23 tRNA genes, and two rRNA genes (Table 2). Individual genes were close together overall and exhibited base overlap, although the exact degree of spacing varied. AT content was higher than GC content, with base compositions exhibiting A+T bias as well as a strong preference for A and C bases. The mtDNA of all eight Carangidae species showed positive AT skewness (0.020–0.060) and negative

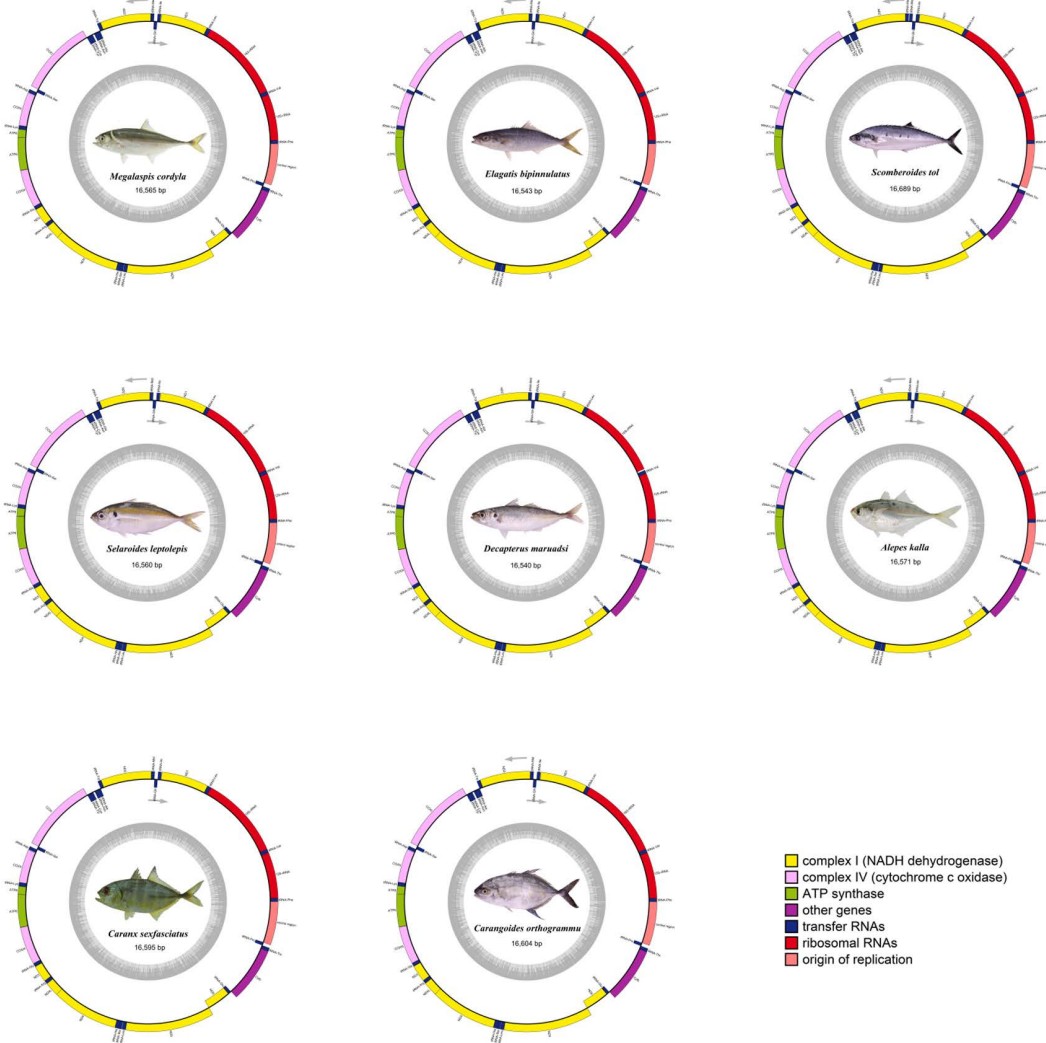

**Fig 1. Mitochondrial genomes map of eight Carangidae species sequenced in this study.**

**Table 2.** Sequence characteristics of mitochondrial genome. + and − correspond to H and L strands, respectively.

| Gene/Species | | From | to | Size | Strand | Nr. of Ami-nao Acids | Anti-Coden | Inferred Termi-nation Coden | GC Percent | AT Percent | Intergenic nucleotides |
|---|---|---|---|---|---|---|---|---|---|---|---|
| tRNA-Phe | M. cordyla | 1 | 68 | 68 | H | GAA | | | 50. 00% | 50.00% | 0 |
| | E. bipinnulatus | 1 | 68 | 68 | H | GAA | | | 42.65% | 57.35% | 0 |
| 12S-rRNA | M. cordyla | 69 | 1022 | 954 | H | | | | 48.53% | 51.47% | 0 |
| | E. bipinnulatus | 69 | 1016 | 954 | H | | | | 47.89% | 52.10% | 0 |
| tRNA-Val | M. cordyla | 1023 | 1094 | 72 | H | TAC | | | 44. 44% | 55.56% | 40 |
| | E. bipinnulatus | 1017 | 1088 | 72 | H | TAC | | | 47. 22% | 52.78% | 39 |
| 16S-rRNA | M. cordyla | 1135 | 2817 | 1683 | H | | | | 45.28% | 54.72% | 0 |
| | E. bipinnulatus | 1128 | 2804 | 1677 | H | | | | 46.15% | 53.85% | 0 |
| tRNA-Leu | M. cordyla | 2818 | 2892 | 75 | H | TAA | | | 49.33% | 50.67% | 0 |
| | E. bipinnulatus | 2805 | 2879 | 75 | H | TAA | | | 46.67% | 53.33% | 0 |
| ND1 | M. cordyla | 2893 | 3867 | 975 | H | | ATG | TAA | 47.59% | 52.41% | 6 |
| | E. bipinnulatus | 2880 | 3854 | 975 | H | | ATG | TAA | 45.95% | 54.05% | 4 |
| tRNA-Ile | M. cordyla | 3874 | 3942 | 69 | H | GAT | | | 53.62% | 46.38% | 0 |
| | E. bipinnulatus | 3859 | 3928 | 70 | H | GAT | | | 48.57% | 51.43% | −1 |
| tRNA-Gln | M. cordyla | 3943 | 4013 | 71 | L | TTG | | | 40.85% | 59.15% | −1 |
| | E. bipinnulatus | 3928 | 3998 | 71 | L | TTG | | | 40.85% | 59.15% | −1 |
| tRNA-Met | M. cordyla | 4013 | 4082 | 70 | H | CAT | | | 51. 43% | 48.57% | 0 |
| | E. bipinnulatus | 3998 | 4066 | 69 | H | CAT | | | 49.28% | 50.72% | 0 |
| ND2 | M. cordyla | 4083 | 5129 | 1047 | H | | ATG | TAG | 46. 42% | 53.58% | −2 |
| | E. bipinnulatus | 4067 | 5113 | 1047 | H | | ATG | TAA | 49.09% | 50.91% | −1 |
| tRNA-Trp | M. cordyla | 5128 | 5198 | 71 | H | TCA | | | 49.30% | 50.70% | 1 |
| | E. bipinnulatus | 5113 | 5183 | 71 | H | TCA | | | 50.70% | 49.30% | 1 |
| tRNA-Ala | M. cordyla | 5200 | 5268 | 69 | L | TGC | | | 40.58% | 59.42% | 1 |
| | E. bipinnulatus | 5185 | 5253 | 69 | L | TGC | | | 40.58% | 59.42% | 1 |
| tRNA-Asn | M. cordyla | 5270 | 5342 | 73 | L | GTT | | | 47. 95% | 52.05% | 37 |
| | E. bipinnulatus | 5255 | 5327 | 73 | L | GTT | | | 49.32% | 50.68% | 39 |
| tRNA-Cys | M. cordyla | 5380 | 5447 | 68 | L | GCA | | | 45.59% | 54.41% | 0 |
| | E. bipinnulatus | 5367 | 5433 | 67 | L | GCA | | | 44.78% | 55.22% | 0 |
| tRNA-Tyr | M. cordyla | 5448 | 5517 | 70 | L | GTA | | | 48.57% | 51.43% | 1 |
| | E. bipinnulatus | 5434 | 5503 | 70 | L | GTA | | | 51.43% | 48.57% | 1 |
| COXI | M. cordyla | 5519 | 7069 | 1551 | H | | GTG | TAA | 44.04% | 55.96% | 0 |
| | E. bipinnulatus | 5505 | 7055 | 1551 | H | | GTG | TAA | 46.87% | 53.13% | 0 |
| tRNA-Ser | M. cordyla | 7070 | 7140 | 71 | L | TGA | | | 47.89% | 52.11% | 3 |
| | E. bipinnulatus | 7056 | 7126 | 71 | L | TGA | | | 47.89% | 52.11% | 3 |
| tRNA-Asp | M. cordyla | 7144 | 7214 | 71 | H | GTC | | | 49.30% | 50.70% | 6 |
| | E. bipinnulatus | 7130 | 7200 | 71 | H | GTC | | | 46.48% | 53.52% | 8 |
| COXII | M. cordyla | 7221 | 7911 | 691 | H | | ATG | T | 43. 27% | 56.73% | 0 |
| | E. bipinnulatus | 7209 | 7899 | 691 | H | | ATG | T | 43.96% | 56.04% | 0 |
| tRNA-Lys | M. cordyla | 7912 | 7986 | 75 | H | TTT | | | 46.67% | 53.33% | 1 |
| | E. bipinnulatus | 7900 | 7974 | 75 | H | TTT | | | 41.33% | 58.67% | 1 |
| ATP8 | M. cordyla | 7988 | 8155 | 168 | H | | ATG | TAA | 44.64% | 55.36% | −4 |
| | E. bipinnulatus | 7976 | 8143 | 168 | H | | ATG | TAA | 44. 05% | 55.95% | −10 |
| ATP6 | M. cordyla | 8152 | 8829 | 678 | H | | ATA | TAA | 44.10% | 55.90% | −1 |
| | E. bipinnulatus | 8134 | 8817 | 684 | H | | ATG | TAA | 45. 32% | 54.68% | −1 |

*(Continued)*

**Table 2.** (Continued)

| Gene/Species | | From | to | Size | Strand | Nr. of Ami-nao Acids | Anti-Coden | Inferred Termi-nation Coden | GC Percent | AT Percent | Intergenic nucleotides |
|---|---|---|---|---|---|---|---|---|---|---|---|
| COXIII | M. cordyla | 8829 | 9614 | 786 | H | | ATG | TAA | 48. 98% | 51.02% | −1 |
| | E. bipinnulatus | 8817 | 9602 | 786 | H | | ATG | TAA | 48. 73% | 51.27% | −1 |
| tRNA-Gly | M. cordyla | 9614 | 9683 | 70 | H | TCC | | | 32. 86% | 67.14% | 0 |
| | E. bipinnulatus | 9602 | 9672 | 71 | H | TCC | | | 36. 62% | 63.38% | 0 |
| ND3 | M. cordyla | 9684 | 10034 | 351 | H | | ATG | TAG | 47. 29% | 52.71% | −2 |
| | E. bipinnulatus | 9673 | 10023 | 351 | H | | ATG | TAG | 49. 57% | 50.43% | −2 |
| tRNA-Arg | M. cordyla | 10033 | 10101 | 69 | H | TCG | | | 26. 09% | 73.91% | 1 |
| | E. bipinnulatus | 10022 | 10090 | 69 | H | TCG | | | 33. 33% | 66.67% | 0 |
| ND4L | M. cordyla | 10103 | 10399 | 297 | H | | ATG | TAA | 51. 52% | 48.48% | −7 |
| | E. bipinnulatus | 10091 | 10387 | 297 | H | | ATG | TAA | 49. 83% | 50.17% | −7 |
| ND4 | M. cordyla | 10393 | 11763 | 1381 | H | | ATG | T | 45. 40% | 54.60% | 0 |
| | E. bipinnulatus | 10381 | 11761 | 1381 | H | | ATG | T | 46. 85% | 53.15% | 0 |
| tRNA-His | M. cordyla | 11774 | 11844 | 71 | H | GTG | | | 38. 03% | 61.97% | 0 |
| | E. bipinnulatus | 11762 | 11830 | 69 | H | GTG | | | 33. 33% | 66.67% | 0 |
| tRNA-Ser | M. cordyla | 11845 | 11912 | 68 | H | GCT | | | 50. 00% | 50.00% | 4 |
| | E. bipinnulatus | 11831 | 11897 | 67 | H | GCT | | | 55. 22% | 44.78% | 4 |
| tRNA-Leu | M. cordyla | 11917 | 11989 | 73 | H | TAG | | | 45. 21% | 54.79% | 0 |
| | E. bipinnulatus | 11902 | 11974 | 73 | H | TAG | | | 45. 21% | 54.79% | 0 |
| ND5 | M. cordyla | 11990 | 13828 | 1839 | H | | ATG | TAA | 44. 05% | 55.95% | −4 |
| | E. bipinnulatus | 11975 | 13813 | 1839 | H | | ATG | TAG | 45. 62% | 54.38% | −4 |
| | C. sexfasciatus | 11988 | 13826 | 1839 | H | | ATG | TAA | 44. 54% | 55.46% | −4 |
| ND6 | M. cordyla | 13825 | 14346 | 522 | L | | ATG | TAG | 43. 49% | 56.51% | 0 |
| | E. bipinnulatus | 13810 | 14331 | 522 | L | | ATG | TAG | 45. 59% | 54.41% | 0 |
| tRNA-Glu | M. cordyla | 14347 | 14415 | 69 | L | TTC | | | 42. 03% | 57.97% | 4 |
| | E. bipinnulatus | 14332 | 14400 | 69 | L | TTC | | | 42. 03% | 57.97% | 4 |
| Cytb | M. cordyla | 14420 | 15560 | 1141 | H | | ATG | T | 47. 06% | 52.94% | 0 |
| | E. bipinnulatus | 14405 | 15545 | 1141 | H | | ATG | T | 47. 50% | 52.50% | 0 |
| tRNA-Thr | M. cordyla | 15561 | 15632 | 72 | H | TGT | | | 55. 56% | 44.44% | −1 |
| | E. bipinnulatus | 15546 | 15617 | 72 | H | TGT | | | 56. 94% | 43.06% | −1 |
| tRNA-Pro | M. cordyla | 15632 | 15702 | 71 | L | TGG | | | 39. 44% | 60.56% | 0 |
| | E. bipinnulatus | 15617 | 15687 | 71 | L | TGG | | | 43. 66% | 56.34% | 0 |
| D-loop | M. cordyla | 15703 | 16565 | 863 | H | | | | 37. 54% | 62.46% | 0 |
| | E. bipinnulatus | 15688 | 16543 | 856 | H | | | | 38. 32% | 61.68% | 0 |

GC-skewness (−0.330–0.265), consistent with the pattern of nucleotide skewness observed in other vertebrate mitochon-drial genomes such as *Scylla paramamosain*, *Arius maculatus* and *Channa siamensis* [37–39].

The mtDNA PCGs of the eight Carangidae species and of *L. calcarifer* encoded 3795–3815 amino acids. The codon usage of PCGs reflected their higher A+T levels and nucleotide skewness, specifically the presence of 37, 40, 39, 40, 36, 39, 38 and 39 preferred codons (RSCU ≥ 1) in 13 PCGs of *M. cordyla, E. bipinnulatus, S. tol, S. leptolepis, D. maruadsi, A. kleinii, C. sexfasciatus,* and *C. orthogrammu* [40]. These codon patterns are remarkably similar to those of other Carangi-dae species, with Leu being the most common used and Stp being the least. Other common codons were those encoding amino acids Ala, Thr, Val, Ser, Pro, Gly, and Ile (Fig 2). Codon distribution and amino acid content corresponded between all nine included species, suggesting amino acid conservation (Fig 3). In addition, codons with A or C in the third position

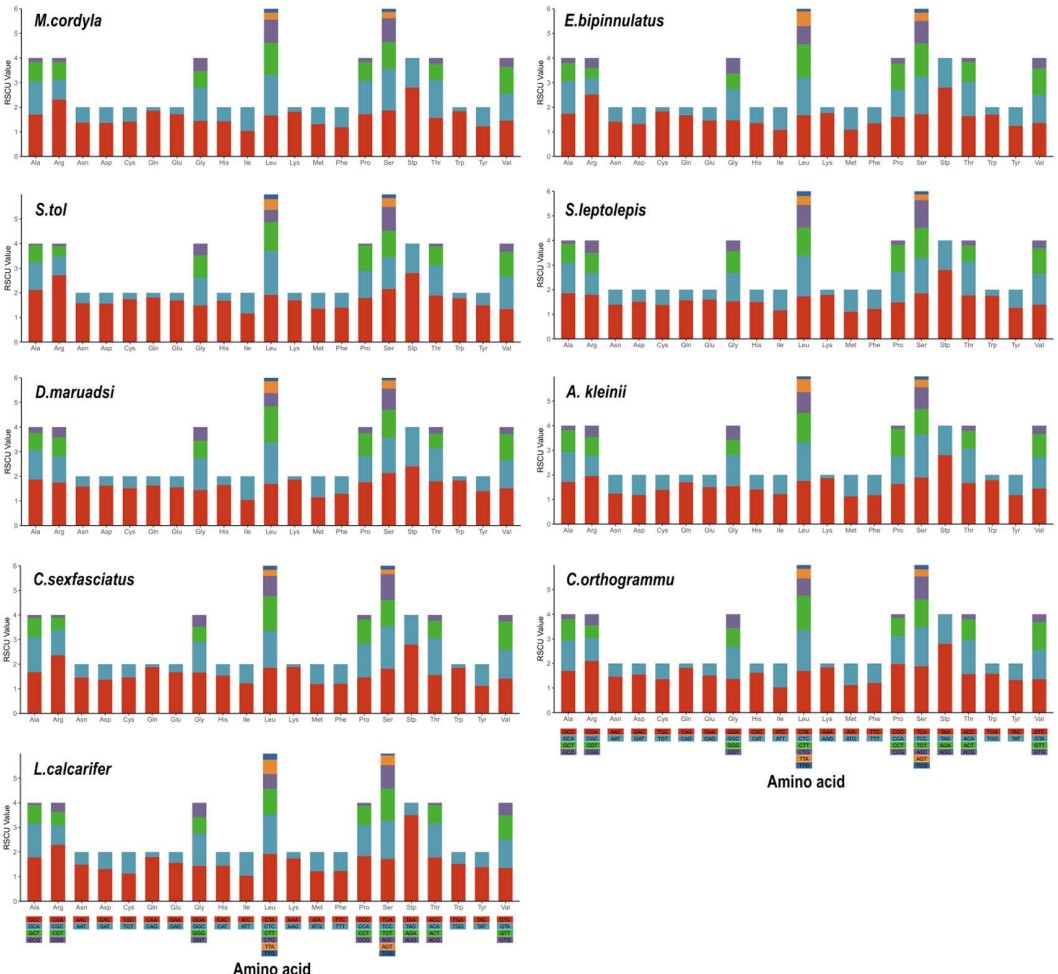

**Fig 2. Relative Synonymous Codon Usage (RSCU) of mitochondrial whole genomes across eight Carangidae species and L. calcarifer.**

were overrepresented compared with synonymous codons. For example, ATG and TCT were rare, whereas the synonymous GCC and GAA were prevalent (Fig 4).

### 3.2 Interspecific genetic distance

*Uraspis secunda* had the smallest genetic distance of 0, while *Uraspis helvola* and *Caranx equula* exhibited the largest genetic distance (0.232) from *Scomberoides lysan*. Mean genetic distance between the 41 species was 0.157. Genetic distances between *Alepes kleinii* was closer to *Alepes djedaba* (0.082) than to *Atule mate* (0.107), while the genetic distance between *A. mate* and *A. djedaba* was 1.104, supporting the categorization of *A. kleinii* in genus *Alepes*. Additionally, *C. equula* had the closest genetic distance (0.111) to *Pseudocaranx dentex*, in comparison to its distance from other Carangoides species: *Carangoides malabaricus* (0.150), *Carangoides plagiotaenia* (0.150), *Carangoides armatus* (0.150), *C. orthogrammu* (0.147), and *Carangoides bajad* (0.146). At 0.099, *C. orthogrammu* had the closest genetic distance to *Parastromateus niger*, followed by *U. helvola* (0.106) (Tables 3, 4). The genetic distances for *C. equula* and *C. orthogrammu* did not conform to their traditional taxonomic categories.

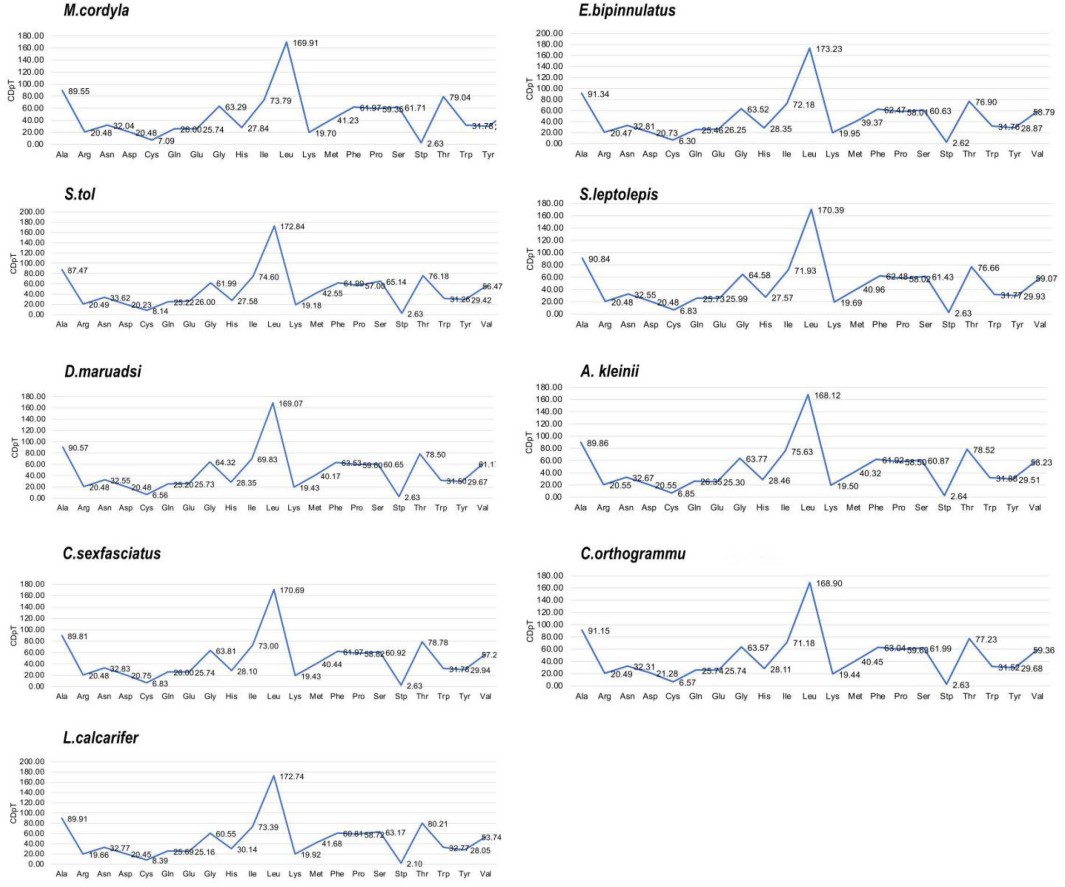

**Fig 3. Codon distribution in members of 19 fish species.** CDspT = codons per thousand codons.

### 3.3 Gene arrangement

Gene rearrangement analysis revealed that *S. tol* and *Scomberoides lysan* experienced the same change, involving an extra tRNA-Met gene between the tRNA-Met and ND2 genes (Fig 5). The mtDNA of the remaining 39 species had identical gene orders, reflecting mtDNA stability in Carangidae. Based on Lü's explanation of mtDNA rearrangement in *Ophichthus brevicaudatus* [41], we can infer that the rearrangement phenomenon in *S. tol* and *S. lysan* can be attributed to random gene duplication. Thus, *Scomberoides* fish can be identified through the presence of duplicate tRNA-Met structures in mtDNA sequences.

### 3.4 Phylogenetic and molecular clock analyses

All three (NJ, ML, and BI) phylogenetic trees yielded similar results, and the majority of their branch bootstrap values were 90–100%, suggesting that the taxonomic relationships were reliable. The phylogenetic analyses indicated that Carangidae comprises two major clades: Chorineminae and Trachinotinae. These two groups combined the earlier isolated clades 1 and 2 of Seriolinae and Caranginae. Seriolinae was the earliest to evolve, followed by Caranginae and Trachinotinae. Chorineminae was the last to evolve and thus exhibits the greatest evolutionary distance and genetic variation. Chorineminae is closely related to Trachinotinae, while Seriolinae is more closely related to Caranginae. *Alepes djedaba* and *A. kleinii* clustered in the same genus within Carangidae, in agreement with the genetic distance results. In contrast

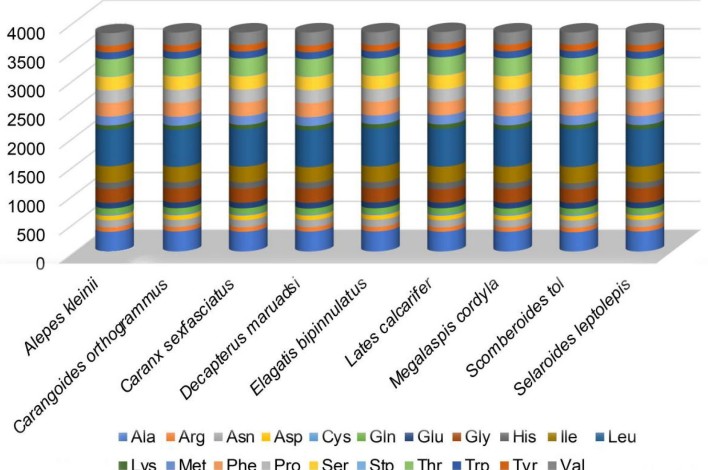

**Fig 4. Codon usage comparison within mitochondrial whole genomes of the nine included fish species.**

**Table 3. Carangidae species used for calculating Kimura 2-parameter pairwise distances based on mitochondrial genomes.**

| No. | Latin name | No. | Latin name | No. | Latin name |
|---|---|---|---|---|---|
| 1 | *Alectis ciliaris* | 15 | *Alectis indica* | 29 | *Pseudocaranx dentex* |
| 2 | *Uraspis helvola* | 16 | *Carangoides plagiotaenia* | 30 | *Trachinotus blochii* |
| 3 | *Decapterus macrosoma* | 17 | *Decapterus macarellus* | 31 | *Caranx ignobilis* |
| 4 | *Alepes kleinii* | 18 | *Trachinotus ovatus* | 32 | *Caranx melampygus* |
| 5 | *Gnathanodon speciosus* | 19 | *Caranx tille* | 33 | *Seriola lalandi* |
| 6 | *Sco Mberoides lysan* | 20 | *Decapterus russelli* | 34 | *Carangoides equula* |
| 7 | *Parastromateus niger* | 21 | *Seriola dumerili* | 35 | *Carangoides armatus* |
| 8 | *Trachurus japonicus* | 22 | *Alepes djedaba* | 36 | *Selaroides leptolepis* |
| 9 | *Trachurus trachuru* | 23 | *Decapterus tabl* | 37 | *Seriolina nigrofasciata* |
| 10 | *Seriola rivoliana* | 24 | *Selar crumenophthalmus* | 38 | *Caranx sexfasciatus* |
| 11 | *Trachinotus carolinus* | 25 | *Seriola quinqueradiata* | 39 | *Sco Mberoides tol* |
| 12 | *Uraspis secunda* | 26 | *Megalaspis cordyla* | 40 | *Carangoides bajad* |
| 13 | *Elagatis bipinnulatus* | 27 | *Atule mate* | 41 | *Carangoides orthogrammus* |
| 14 | *Carangoides malabaricus* | 28 | *Decapterus maruadsi* | 29 | *Pseudocaranx dentex* |

with traditional classification, *C. equula* clustered with *Pseudocaranx dentex* in a group that is more closely related to Trachurus and Decapterus. Similarly differing from traditional classification, *C. orthogrammu* clustered with *Uraspis secunda* and *U. helvola* in a group more closely related to *P. niger* (Figs 6–8).

The Carangidae phylogenetic tree was consistent with the molecular clock analysis and had a high confidence level (Fig 9). Using the divergence times of *Alectis ciliaris* and *A. indicus* as reference divergence times, we observed that Carangidae and *L. calcarifer* shared a common origin in the early Cenomanian of the Late Cretaceous, approximately 99.66 Mya. Chorineminae and Trachinotinae co-originated in the Alternate Late Cretaceous and Paleocene, approximately 68.5 Mya, whereas Seriolinae and Caranginae co-originated in the Selenite phase of the Paleocene, approximately 62.25 Mya. From the middle Eocene to the late Pleistocene (~40.34–0.15 Mya), an explosive divergence led to the formation of *Decapterus, Trachurus, Pseudocaranx, Selar, Caranx, Carangoides,* and *Trachinotus*. The earliest and latest genera to diverge were *Scomberoides* at approximately 46.80 Mya and *Uraspis* at approximately 0.15 Mya.

**Table 4. Kimura-2-parameter pairwise distances of the subfamily Carangidae based on mitochondrial whole genomes.**

| | 1 | 2 | 3 | 4 | 5 | 6 | 7 | 8 | 9 | 10 | 11 | 12 | 13 | 14 | 15 | 16 | 17 | 18 | 19 | 20 | 21 | 22 |
|---|---|---|---|---|---|---|---|---|---|---|---|---|---|---|---|---|---|---|---|---|---|---|
| 2 | 0.117 | | | | | | | | | | | | | | | | | | | | | |
| 3 | 0.142 | 0.135 | | | | | | | | | | | | | | | | | | | | |
| 4 | 0.140 | 0.133 | 0.147 | | | | | | | | | | | | | | | | | | | |
| 5 | 0.131 | 0.130 | 0.142 | 0.126 | | | | | | | | | | | | | | | | | | |
| 6 | 0.226 | 0.220 | 0.222 | 0.225 | 0.223 | | | | | | | | | | | | | | | | | |
| 7 | 0.111 | 0.101 | 0.132 | 0.135 | 0.127 | 0.218 | | | | | | | | | | | | | | | | |
| 8 | 0.133 | 0.129 | 0.107 | 0.143 | 0.138 | 0.220 | 0.125 | | | | | | | | | | | | | | | |
| 9 | 0.132 | 0.128 | 0.105 | 0.142 | 0.138 | 0.222 | 0.125 | 0.027 | | | | | | | | | | | | | | |
| 10 | 0.176 | 0.174 | 0.174 | 0.182 | 0.175 | 0.224 | 0.170 | 0.174 | 0.174 | | | | | | | | | | | | | |
| 11 | 0.185 | 0.189 | 0.190 | 0.191 | 0.188 | 0.208 | 0.185 | 0.184 | 0.186 | 0.187 | | | | | | | | | | | | |
| 12 | 0.117 | 0.002 | 0.134 | 0.133 | 0.130 | 0.220 | 0.101 | 0.128 | 0.127 | 0.174 | 0.189 | | | | | | | | | | | |
| 13 | 0.165 | 0.160 | 0.165 | 0.172 | 0.163 | 0.211 | 0.159 | 0.161 | 0.161 | 0.153 | 0.179 | 0.159 | | | | | | | | | | |
| 14 | 0.118 | 0.119 | 0.136 | 0.141 | 0.136 | 0.222 | 0.115 | 0.132 | 0.129 | 0.177 | 0.191 | 0.118 | 0.160 | | | | | | | | | |
| 15 | 0.113 | 0.120 | 0.143 | 0.142 | 0.137 | 0.220 | 0.116 | 0.135 | 0.135 | 0.177 | 0.189 | 0.119 | 0.166 | 0.122 | | | | | | | | |
| 16 | 0.120 | 0.121 | 0.141 | 0.136 | 0.133 | 0.222 | 0.117 | 0.132 | 0.131 | 0.176 | 0.188 | 0.120 | 0.163 | 0.108 | 0.124 | | | | | | | |
| 17 | 0.140 | 0.133 | 0.073 | 0.148 | 0.145 | 0.221 | 0.131 | 0.099 | 0.098 | 0.175 | 0.189 | 0.133 | 0.164 | 0.137 | 0.141 | 0.139 | | | | | | |
| 18 | 0.186 | 0.185 | 0.189 | 0.189 | 0.189 | 0.223 | 0.182 | 0.184 | 0.184 | 0.187 | 0.079 | 0.185 | 0.178 | 0.188 | 0.189 | 0.183 | 0.190 | | | | | |
| 19 | 0.129 | 0.128 | 0.142 | 0.124 | 0.118 | 0.223 | 0.123 | 0.133 | 0.134 | 0.177 | 0.191 | 0.128 | 0.169 | 0.132 | 0.135 | 0.131 | 0.141 | 0.189 | | | | |
| 20 | 0.142 | 0.135 | 0.081 | 0.147 | 0.144 | 0.220 | 0.133 | 0.102 | 0.101 | 0.178 | 0.189 | 0.134 | 0.164 | 0.137 | 0.144 | 0.138 | 0.073 | 0.190 | 0.141 | | | |
| 21 | 0.175 | 0.175 | 0.176 | 0.180 | 0.177 | 0.225 | 0.173 | 0.176 | 0.176 | 0.059 | 0.189 | 0.175 | 0.156 | 0.174 | 0.178 | 0.175 | 0.177 | 0.192 | 0.177 | 0.179 | | |
| 22 | 0.137 | 0.132 | 0.145 | 0.082 | 0.129 | 0.227 | 0.135 | 0.140 | 0.139 | 0.181 | 0.192 | 0.132 | 0.170 | 0.139 | 0.142 | 0.137 | 0.144 | 0.191 | 0.126 | 0.146 | 0.182 | |
| 23 | 0.141 | 0.133 | 0.096 | 0.143 | 0.142 | 0.220 | 0.131 | 0.095 | 0.095 | 0.173 | 0.187 | 0.132 | 0.163 | 0.138 | 0.141 | 0.138 | 0.093 | 0.188 | 0.140 | 0.093 | 0.173 | 0.144 |

*(Continued)*

**Table 4.** (Continued)

| | 1 | 2 | 3 | 4 | 5 | 6 | 7 | 8 | 9 | 10 | 11 | 12 | 13 | 14 | 15 | 16 | 17 | 18 | 19 | 20 | 21 | 22 | 23 | 24 | 25 | 26 | 27 | 28 | 29 | 30 | 31 | 32 | 33 | 34 | 35 | 36 | 37 | 38 | 39 | 40 | 41 |
|---|---|---|---|---|---|---|---|---|---|---|---|---|---|---|---|---|---|---|---|---|---|---|---|---|---|---|---|---|---|---|---|---|---|---|---|---|---|---|---|---|---|
| 24 | 0.152 | 0.145 | 0.147 | 0.155 | 0.153 | 0.225 | 0.148 | 0.141 | 0.141 | 0.184 | 0.196 | 0.145 | 0.172 | 0.151 | 0.153 | 0.147 | 0.146 | 0.196 | 0.150 | 0.148 | 0.181 | 0.156 | 0.144 | | | | | | | | | | | | | | | | | | |
| 25 | 0.181 | 0.176 | 0.178 | 0.183 | 0.180 | 0.227 | 0.173 | 0.177 | 0.176 | 0.090 | 0.191 | 0.176 | 0.160 | 0.180 | 0.179 | 0.179 | 0.177 | 0.195 | 0.180 | 0.179 | 0.090 | 0.185 | 0.175 | 0.185 | | | | | | | | | | | | | | | | | |
| 26 | 0.130 | 0.130 | 0.146 | 0.127 | 0.119 | 0.225 | 0.125 | 0.137 | 0.136 | 0.181 | 0.190 | 0.130 | 0.164 | 0.135 | 0.136 | 0.132 | 0.145 | 0.190 | 0.095 | 0.143 | 0.181 | 0.129 | 0.140 | 0.156 | 0.184 | | | | | | | | | | | | | | | | |
| 27 | 0.140 | 0.137 | 0.148 | 0.107 | 0.130 | 0.229 | 0.136 | 0.141 | 0.141 | 0.184 | 0.194 | 0.136 | 0.172 | 0.140 | 0.149 | 0.139 | 0.149 | 0.192 | 0.127 | 0.149 | 0.184 | 0.104 | 0.144 | 0.158 | 0.187 | 0.130 | | | | | | | | | | | | | | | |
| 28 | 0.139 | 0.132 | 0.078 | 0.144 | 0.143 | 0.218 | 0.131 | 0.098 | 0.097 | 0.176 | 0.188 | 0.131 | 0.162 | 0.136 | 0.139 | 0.138 | 0.068 | 0.187 | 0.138 | 0.031 | 0.178 | 0.144 | 0.089 | 0.146 | 0.176 | 0.142 | 0.147 | | | | | | | | | | | | | | |
| 29 | 0.148 | 0.141 | 0.140 | 0.155 | 0.151 | 0.225 | 0.139 | 0.133 | 0.133 | 0.184 | 0.198 | 0.141 | 0.175 | 0.149 | 0.147 | 0.150 | 0.140 | 0.198 | 0.145 | 0.140 | 0.187 | 0.152 | 0.135 | 0.157 | 0.186 | 0.148 | 0.158 | 0.136 | | | | | | | | | | | | | |
| 30 | 0.185 | 0.185 | 0.187 | 0.189 | 0.186 | 0.208 | 0.181 | 0.183 | 0.182 | 0.187 | 0.080 | 0.185 | 0.175 | 0.186 | 0.188 | 0.183 | 0.186 | 0.061 | 0.189 | 0.186 | 0.189 | 0.189 | 0.186 | 0.193 | 0.193 | 0.188 | 0.192 | 0.185 | 0.196 | | | | | | | | | | | | |
| 31 | 0.132 | 0.133 | 0.145 | 0.130 | 0.119 | 0.226 | 0.127 | 0.139 | 0.137 | 0.182 | 0.195 | 0.133 | 0.170 | 0.138 | 0.138 | 0.134 | 0.146 | 0.194 | 0.076 | 0.145 | 0.181 | 0.132 | 0.142 | 0.155 | 0.182 | 0.100 | 0.131 | 0.141 | 0.149 | 0.190 | | | | | | | | | | | |
| 32 | 0.132 | 0.135 | 0.146 | 0.129 | 0.120 | 0.224 | 0.126 | 0.137 | 0.136 | 0.180 | 0.193 | 0.134 | 0.169 | 0.137 | 0.139 | 0.136 | 0.146 | 0.190 | 0.070 | 0.143 | 0.181 | 0.131 | 0.145 | 0.154 | 0.181 | 0.098 | 0.130 | 0.142 | 0.149 | 0.189 | 0.078 | | | | | | | | | | |
| 33 | 0.179 | 0.172 | 0.177 | 0.183 | 0.179 | 0.230 | 0.175 | 0.175 | 0.176 | 0.085 | 0.191 | 0.171 | 0.158 | 0.179 | 0.179 | 0.179 | 0.176 | 0.195 | 0.181 | 0.179 | 0.084 | 0.175 | 0.175 | 0.184 | 0.055 | 0.184 | 0.185 | 0.176 | 0.186 | 0.192 | 0.183 | 0.184 | | | | | | | | | |
| 34 | 0.152 | 0.146 | 0.144 | 0.159 | 0.156 | 0.232 | 0.142 | 0.136 | 0.133 | 0.188 | 0.202 | 0.147 | 0.178 | 0.150 | 0.153 | 0.150 | 0.142 | 0.203 | 0.151 | 0.142 | 0.189 | 0.158 | 0.137 | 0.158 | 0.191 | 0.153 | 0.163 | 0.141 | 0.111 | 0.202 | 0.156 | 0.156 | 0.191 | | | | | | | | |
| 35 | 0.117 | 0.121 | 0.139 | 0.137 | 0.135 | 0.219 | 0.113 | 0.133 | 0.132 | 0.174 | 0.186 | 0.120 | 0.160 | 0.108 | 0.118 | 0.105 | 0.141 | 0.184 | 0.132 | 0.142 | 0.176 | 0.139 | 0.137 | 0.151 | 0.180 | 0.131 | 0.139 | 0.140 | 0.146 | 0.182 | 0.135 | 0.134 | 0.178 | 0.150 | | | | | | | |
| 36 | 0.147 | 0.140 | 0.149 | 0.139 | 0.135 | 0.229 | 0.138 | 0.146 | 0.147 | 0.182 | 0.192 | 0.140 | 0.172 | 0.145 | 0.146 | 0.142 | 0.149 | 0.191 | 0.133 | 0.148 | 0.181 | 0.141 | 0.147 | 0.158 | 0.183 | 0.136 | 0.143 | 0.148 | 0.161 | 0.190 | 0.141 | 0.138 | 0.181 | 0.165 | 0.144 | | | | | | |
| 37 | 0.182 | 0.177 | 0.179 | 0.183 | 0.178 | 0.227 | 0.177 | 0.177 | 0.179 | 0.095 | 0.193 | 0.177 | 0.160 | 0.179 | 0.184 | 0.180 | 0.181 | 0.194 | 0.183 | 0.180 | 0.087 | 0.184 | 0.179 | 0.186 | 0.102 | 0.184 | 0.187 | 0.179 | 0.191 | 0.195 | 0.184 | 0.185 | 0.093 | 0.195 | 0.175 | 0.184 | | | | | |
| 38 | 0.129 | 0.127 | 0.142 | 0.123 | 0.117 | 0.223 | 0.122 | 0.133 | 0.134 | 0.177 | 0.191 | 0.128 | 0.168 | 0.131 | 0.135 | 0.131 | 0.141 | 0.189 | 0.003 | 0.140 | 0.177 | 0.126 | 0.140 | 0.149 | 0.180 | 0.094 | 0.127 | 0.138 | 0.145 | 0.188 | 0.075 | 0.070 | 0.182 | 0.152 | 0.131 | 0.132 | 0.183 | | | | |
| 39 | 0.221 | 0.215 | 0.216 | 0.225 | 0.217 | 0.094 | 0.217 | 0.216 | 0.218 | 0.224 | 0.206 | 0.214 | 0.208 | 0.217 | 0.220 | 0.218 | 0.219 | 0.205 | 0.218 | 0.217 | 0.224 | 0.223 | 0.216 | 0.222 | 0.224 | 0.218 | 0.224 | 0.216 | 0.225 | 0.206 | 0.222 | 0.221 | 0.225 | 0.231 | 0.220 | 0.222 | 0.223 | 0.218 | | | |
| 40 | 0.112 | 0.111 | 0.135 | 0.132 | 0.127 | 0.219 | 0.107 | 0.128 | 0.128 | 0.171 | 0.184 | 0.111 | 0.158 | 0.100 | 0.112 | 0.091 | 0.133 | 0.181 | 0.124 | 0.134 | 0.171 | 0.133 | 0.133 | 0.145 | 0.175 | 0.126 | 0.134 | 0.134 | 0.142 | 0.179 | 0.130 | 0.128 | 0.174 | 0.146 | 0.094 | 0.138 | 0.176 | 0.123 | 0.216 | | |
| 41 | 0.121 | 0.106 | 0.136 | 0.138 | 0.132 | 0.218 | 0.099 | 0.129 | 0.130 | 0.175 | 0.187 | 0.106 | 0.163 | 0.123 | 0.124 | 0.122 | 0.135 | 0.186 | 0.127 | 0.138 | 0.175 | 0.139 | 0.134 | 0.151 | 0.177 | 0.134 | 0.139 | 0.135 | 0.142 | 0.185 | 0.133 | 0.132 | 0.178 | 0.147 | 0.120 | 0.143 | 0.179 | 0.127 | 0.216 | 0.114 | |

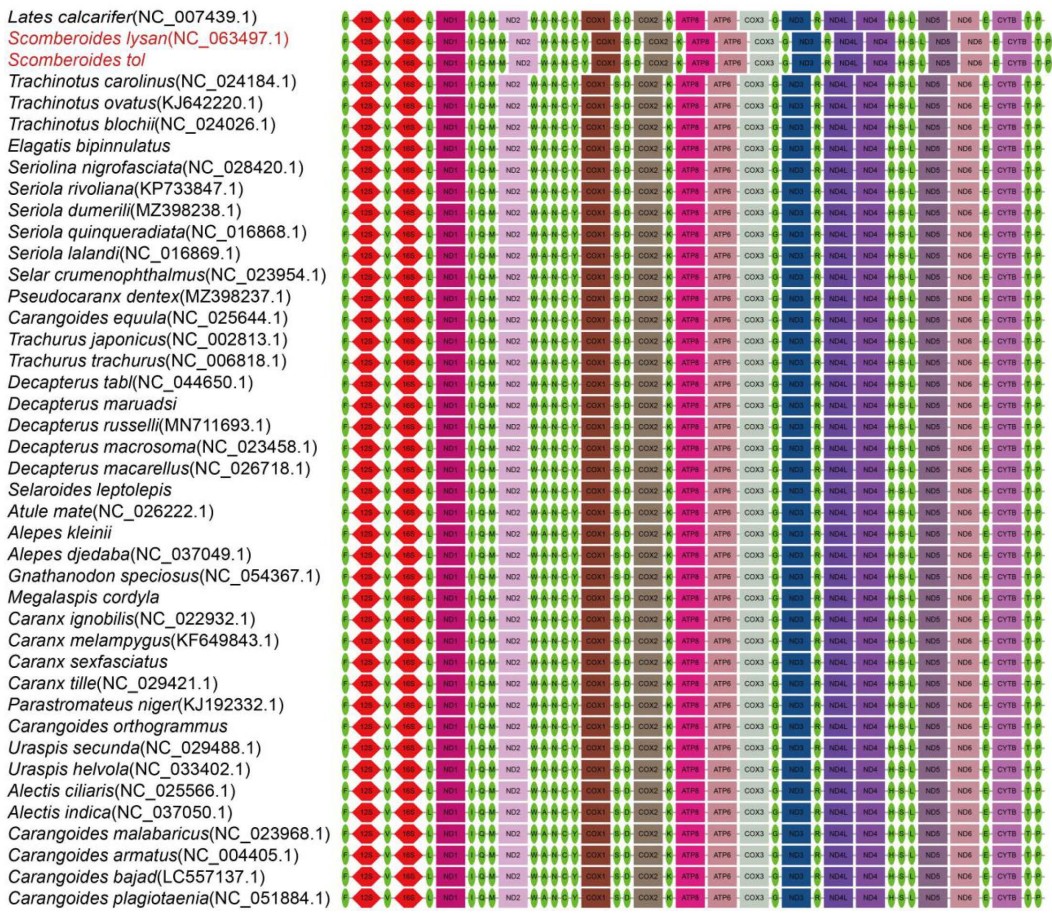

**Fig 5. Mitochondrial whole genome-based gene sequence alignment of 41 Carangidae species.**

The temporal phylogenetic tree of Carangidae allowed us to derive the following evolutionary relationships between the four common subfamilies: Seriolinae diverged earliest, at approximately 46.80 Mya during the mid-Eocene; Caranginae diverged second at approximately 40.34 Mya during the Late Eocene; Chorineminae diverged third at approximately 15.31 Mya during the Early Miocene; and Trachinotinae diverged last at approximately 12.05 Mya during the Late Miocene.

## 4. Discussion

### 4.1 Mitochondrial genomes of the eight Carangidae species

Because mtDNA show distinct and relatively independent matrilineal inheritance, they are suitable for phylogenetic analyses [42,43]. In this study, we sequenced the whole mitochondrial genomes of *S. tol*, *C. orthogrammu*, and *C. sexfasciatus* for the first time. Comparative analysis of mtDNA sequences across eight Carangidae species revealed that their sequence compositions and alignments were essentially the same (except for the presence of two consecutive tRNA-Met genes in *S. tol*), and all showed a clear A+T bias. Their mtDNA sizes ranged from 16540 bp to 16689 bp, with the shortest being *D. maruadsi* and the longest being *S. tol*. All contained 13 PCGs, 22 tRNA genes, two rRNA genes, and a D-loop region, consistent with the structure of other Carangidae genes [44–46]. The one exception was *S. tol*, where 23 tRNA genes were present. AT-skew was positive while GC-skew was negative, typical of nucleotide skew in vertebrate mtDNA [47]. Furthermore, base overlaps and spacers were present, with most occurring at the same locations across all eight species.

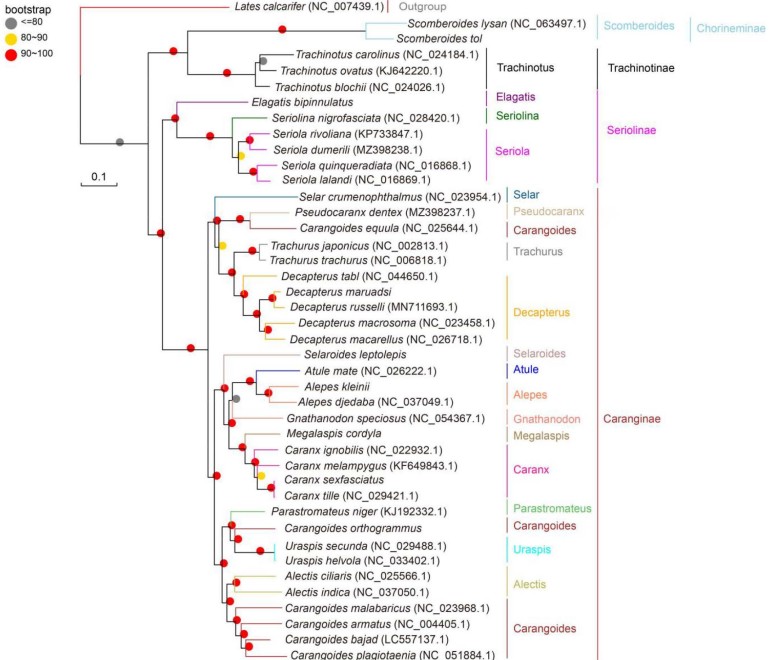

**Fig 6. Maximum-likelihood phylogenetic trees of Carangidae species based on whole mitochondrial genomes.**

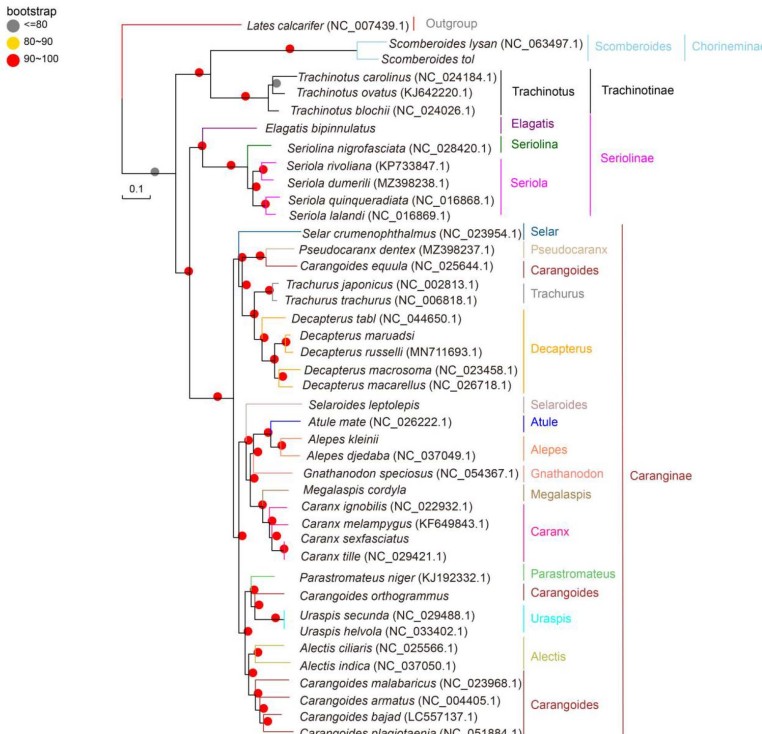

**Fig 7. Bayesian phylogenetic trees (Precentages: 0-100%) of Carangidae species based on complete mitochondrial whole genome.**

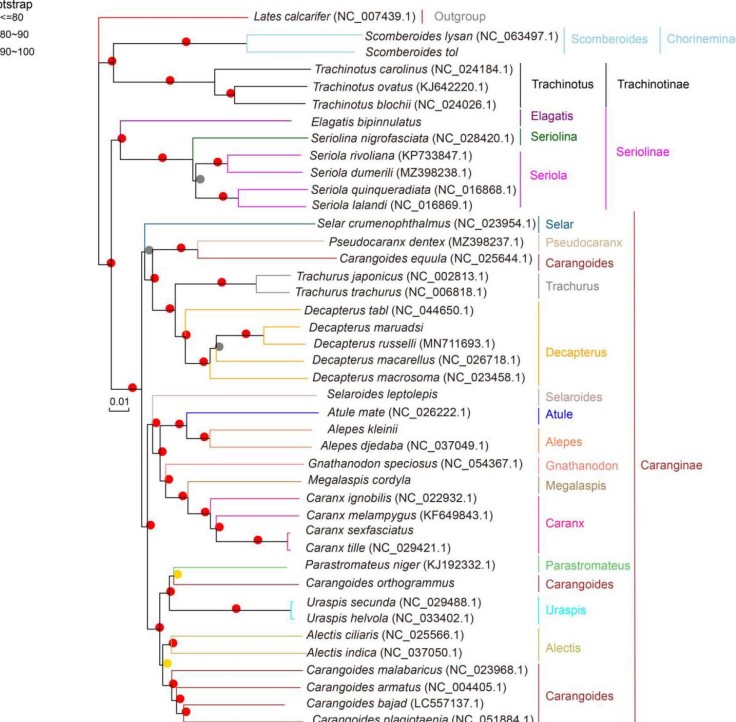

**Fig 8. Neighbor-joining phylogenetic trees of Carangidae species based on whole mitochondrial genome.**

The tRNA genes were dispersed between PCGs, ranging from 1553 bp to 1619 bp in length, with the longest being from *S. tol* (with two tRNA-Met, tRNA-Ser, and tRNA-Leu); the remaining eight species (including the outgroup) had two tRNA-Ser and tRNA-Leu. The 13 PCGs ranged from 11373 bp to 11433 bp in length across the eight Carangidae species, with multiple overlaps; *ND6* was located on the L chain, while the remaining 12 PCGs were located on the H chain. The start codon for all PCGs (except *COXI* with CTG) was ATG. Codon preference analysis showed that the eight Carangidae species encoded 3795–3815 amino acids. Moreover, mtDNA exhibited a A+T bias, while nucleotide skewness was also reflected in PCG codon usage, with 36–40 preferred codons (RSCU ≥ 1) in the 13 PCGs. Codons A and C in the third position were overrepresented.

## 4.2 Phylogeny and divergence

Carangidae phylogeny is controversial, mainly due to unresolved evolutionary relationships of the four subfamilies Carangidae, Seriolinae, Trachinotinae, and Chorineminae, the classification of *A. kleinii* and *C. equula*, as well as potential synonymous species [19,48–50]. The study clarified phylogenetic controversies within the Carangidae family, confirmed that *A. kleinii* and *A. djedaba* belong to the same genus, identified classification issues related to *C. equula* and *C. orthogrammus*. and for the first time estimated the divergence times of Carangidae. Our ML, BI, and NJ phylogenetic trees based on mtDNA of 41 Carangidae species had bootstrap values ranging from 90% to 100%, reflecting reliable outcomes [51]. Our data demonstrated that Carangidae is divided into two major clades, with Chorineminae + Trachinotinae forming one cluster, and Seriolinae + Carangidae forming another. This result is consistent with Gushiken's view that the two pairs are sister groups [52]. A previous study, however, used the mitochondrial genomes of 33 Carangidae species and concluded that Trachinotinae and Seriolinae were sister groups [53]. Other research using mitochondrial cytochrome b to generate three (MP, ML, BI) phylogenetic trees found that Carangidae and Seriolinae clustered together first, then re-clustered with

**Fig 9. Phylogenetic time tree of the Carangidae based on whole mitochondrial genome.**

Trachinotinae, and finally with Chorineminae [54]. A phylogenetic analysis focusing on the mtDNA control region produced a similar result. In our study, evolutionary distance data revealed that Seriolinae evolved the earliest as the original population of Carangidae, whereas Chorineminae diverged the latest and exhibited the greatest genetic variation.

Our species classification clustered *A. djedaba* and *A. kleinii* together into *Alepes*, corroborating the work of Xu [48], based on DNA barcodes of Carangidae in the Putuo Sea, and of Zheng [19], based on 16SrRNA partial sequences. In contrast, a study using morphology grouped *A. kleinii* and *A. djedaba* into *Atule* in the East China Sea Fish Book [55], whereas Liu's List of Marine Organisms of China placed *A. kleinii* in *Atule* and *A. djedaba* in *Alepes* [56]. These contradictions can be attributed to phenotypic variation in response to environmental change across different growth cycles [57], emphasizing the need to combine morphology with molecular analysis to improve taxonomic classification. In this study, we provided strong evidence that *A. kleinii* and *A. djedaba* belonged in the same genus, given a K-2-P distance of 0.082 for both, supporting the scientific name *A. kleinii* rather than *Caranx kalla*. We also noted a genetic distance of 0.002 between *U. helvola* and *U. secunda*, indicating genetic convergence [20]. Furthermore, *C. equula* clustered with *Pseudocaranx dentex* and was more closely related to Trachurus and Decapterus fishes, corroborating the results of Wang [50,53]. However, in contrast with traditional classification, we found that *C. orthogrammus* was more closely related to *U. helvola*, *U. secunda*, and *P. niger*.

The results of molecular clock analysis indicated that the included Carangidae species and *L. calcarifer* shared a common ancestor originating in the early Cenomanian of the Late Cretaceous. Seriolinae differentiated the earliest, during the mid-Eocene, whereas Trachinotinae was the last to differentiate, during the late Miocene. The family then diversified considerably at approximately 40.34 Mya to 0.15 Mya. Finally, we discovered that the presence of two tRNA-Met mutations between tRNA-Gln and ND2 in *S. tol* and *S. lysan* was potentially unique to *Scomberoides*. Thus, future studies should continue sequencing mtDNA from other *Scomberoides* species to verify this gene arrangement and determine whether it is a suitable identifier for the genus.

## 5. Conclusions

This study assembled the mtDNAs of eight Carangidae species, among which the mtDNAs of three species were sequenced for the first time using second-generation sequencing technology and bioinformatics analyses. The remaining five species had their mtDNAs sequenced previously but were included in this study for comparative purposes. Furthermore, we successfully constructed three phylogenetic trees and contributed to clarifying several taxonomic controversies in this family. Through gene arrangement analysis, we identified 23 tRNAs in both *S. tol* and *S. lysan*, with the extra tRNA comprising two repetitive tRNA-Met structures between tRNA-Gln and ND2; this potentially unique signature can be tentatively considered a molecular identifier for *Scomberoides*. Next, phylogenetic analysis revealed that *A. kleinii* clustered with *A. djedaba* in one clade, suggesting that *A. kleinii* belongs to *Alepes*. Additionally, the extremely small genetic distance between *U. secunda* and *U. helvola* implies that the two species may be synonymous, a hypothesis that requires further verification in conjunction with morphological analysis. Both *C. equula* and *C. orthogrammu* had intrageneric distances greater than their intergeneric distances, and phylogenetic analyses demonstrated that they did not cluster with other Carangoides fishes, inconsistent with their traditional taxonomic status. Molecular clock analysis then showed that among the four subfamilies, Seriolinae diverged first, followed by Carangidae, then Chorineminae, and finally Trachinotinae. Moreover, Seriolinae and Carangidae are clustered together in one group, while Trachinotinae and Chorineminae are clustered together in another.

The phylogenetic and taxonomic findings of this study also have important conservation implications, especially for overfished species. The nearly identical genetics of *U.secunda* and *U.helvola*, which might be synonyms, mean their conservation status needs re-evaluation to ensure proper management. Similarly, reclassifying *A.kleinii* and *A.djedaba* into Alepes shows we must reassess their ecological roles and population trends to understand their vulnerability to overfishing and habitat loss. The unexpected phylogenetic positions of *C.equula* and *C.orthogrammus* show traditional taxonomy can't fully reflect species' true distinctiveness, risking insufficient protection. Combining genetic data with ecological and fisheries info can help design better conservation strategies. Also, the unique gene arrangement in *Scomberoides* (two tRNA - Met structures between tRNA - Gln and ND2) can be a molecular marker for species ID. This is vital for monitoring and managing populations, especially those targeted by fisheries, as accurate ID is crucial for sustainable resource use.

In conclusion, this study provides a theoretical basis for further research on taxonomy and evolutionary genetics, while also benefiting the development of improved germplasm resources and the conservation of the fish in Carangidae species.

## Author contributions

**Data curation:** Nan Zhang.

**Investigation:** Huayang Guo.

**Methodology:** Baosuo Liu.

**Project administration:** Dianchang Zhang.

**Software:** Lin Xian.

**Supervision:** Kecheng Zhu.

**Visualization:** Jingwen Yang, Bo Liu.

**Writing – original draft:** Fangcao Zhao.

**Writing – review & editing:** Fangcao Zhao, Lin Xian.

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
