## [Decision Letter · Decision Letter 0]

PONE-D-24-54115Mitochondrial genome assembly of eight Carangidae and phylogenetic analysis of Carangidae familyPLOS ONE

Dear Dr. Zhang,

Thank you for submitting your manuscript to PLOS ONE. After careful consideration, we feel that it has merit but does not fully meet PLOS ONE’s publication criteria as it currently stands. Therefore, we invite you to submit a revised version of the manuscript that addresses the points raised during the review process.

We look forward to receiving your revised manuscript.

Kind regards,

Muhammad Asghar Hassan, Ph.D

Academic Editor

PLOS ONE

Journal requirements: When submitting your revision, we need you to address these additional requirements. 1. Please ensure that your manuscript meets PLOS ONE's style requirements, including those for file naming. The PLOS ONE style templates can be found at https://journals.plos.org/plosone/s/file?id=wjVg/PLOSOne_formatting_sample_main_body.pdf and https://journals.plos.org/plosone/s/file?id=ba62/PLOSOne_formatting_sample_title_authors_affiliations.pdf. 2. To comply with PLOS ONE submissions requirements, in your Methods section, please provide additional information regarding the experiments involving animals and ensure you have included details on (1) methods of sacrifice, (2) methods of anesthesia and/or analgesia, and (3) efforts to alleviate suffering. 3. Thank you for stating the following financial disclosure:  [Central Public-Interest Scientific Institution Basal Research Fund, CAFS (NO. 2022TS07)].  Please state what role the funders took in the study.  If the funders had no role, please state: ""The funders had no role in study design, data collection and analysis, decision to publish, or preparation of the manuscript."" If this statement is not correct you must amend it as needed. Please include this amended Role of Funder statement in your cover letter; we will change the online submission form on your behalf. 4. Thank you for stating the following in the Acknowledgments Section of your manuscript: [This study was supported by Central Public-Interest Scientific Institution Basal Research Fund, CAFS (NO. 2022TS07), National Marine Genetic Resource Center, Financial Fund of the Ministry of Agriculture and Rural Affairs, P. R. of China (NHYYSWZZZYKZX2020) and Operation and Maintenance of Guangdong Marine Aquatic Germplasm Resource Bank (2022-SBH-00-002).]We note that you have provided funding information that is not currently declared in your Funding Statement. However, funding information should not appear in the Acknowledgments section or other areas of your manuscript. We will only publish funding information present in the Funding Statement section of the online submission form. Please remove any funding-related text from the manuscript and let us know how you would like to update your Funding Statement. Currently, your Funding Statement reads as follows:   [Central Public-Interest Scientific Institution Basal Research Fund, CAFS (NO. 2022TS07)].  Please include your amended statements within your cover letter; we will change the online submission form on your behalf. 5. Please include your full ethics statement in the ‘Methods’ section of your manuscript file. In your statement, please include the full name of the IRB or ethics committee who approved or waived your study, as well as whether or not you obtained informed written or verbal consent. If consent was waived for your study, please include this information in your statement as well. 

Reviewers' comments:

Reviewer's Responses to Questions

**Comments to the Author**

1. Is the manuscript technically sound, and do the data support the conclusions?

Reviewer #1: Yes

Reviewer #2: Yes

2. Has the statistical analysis been performed appropriately and rigorously? 

Reviewer #1: Yes

Reviewer #2: Yes

3. Have the authors made all data underlying the findings in their manuscript fully available?

Reviewer #1: Yes

Reviewer #2: Yes

4. Is the manuscript presented in an intelligible fashion and written in standard English?

Reviewer #1: Yes

Reviewer #2: Yes

5. Review Comments to the Author

Reviewer #1: Dear Authors,

I hope this message finds you well.

The author has conducted valuable research on the mitochondrial genome assembly of eight Carangidae species and the phylogenetic analysis of the Carangidae family. While the article is well-written overall, there are some suggestions and corrections needed to enhance its quality.

I kindly request your assistance in improving the overall language quality of the article to enhance its clarity and readability. Additionally, I noticed that the figures provided could benefit from quality improvements to better support the content of the manuscript.

Your expertise in refining the text and upgrading the figures would significantly improve the overall presentation of the article. Below, I have outlined the suggestions and corrections for your consideration:

1). Title: The title needs to be revised. It is not well constructed. I suggest directly mentioning mitochondrial genomes and their phylogenetic analysis. The family name appears twice, and the mention of "assembly" is unnecessary.

2) Line 28: Change "Genome sizes were" to "Mitochondrial genome sizes were."

3). Line 33: Clarify the sentence. Did you mean that these genera were sequenced for the first time or these species? If it is the species, why were the others sequenced again?

4) Introduction, Line 46: Revise this sentence for better clarity.

5) Lines 69–72: Provide relevant citations and specify the results. Are these results based on mitochondrial genomes? Clarify which datasets were used and whether the phylogeny focuses on a species, family, or genus.

6) Line 94: Revise this line for clarity.

7) Table 1: Ensure all Latin names are italicized. Review the entire table for consistency. In the legends, change "Genric" to "Genus."

8) Line 157: Add references to support the statement "nucleotide skewness observed in other vertebrate mitochondrial genomes." Specify which studies or examples you are referring to?.

9) Table 2: Explain why you provided GC percentages but not AT content. Alternatively, consider providing both values for completeness.

10) Table 2 Length: The table is too long. Consider presenting data for one or two species in the main table and moving the rest to supplementary materials.

11) Table 3: This table does not look appropriate in the main text. It should be moved to the supporting materials.

12) Line 224: Revise the sentence "Alepes djedaba and A. kleinii clustered in the same genus within Carangidae." Ensure that species names are italicized.

13) Line 279: You mentioned that new classification problems were identified. Clearly explain these issues and highlight them in your results. This will enhance the value of your discovery and the overall article.

14) Figure 1: Revise the caption to read. Add map. Like "Mitochondrial genome maps of eight..." Also, improve the figure's quality.

15) Figures 6 and 7: It seems like these figures are the same, but you mentioned them as ML and BI analyses. Clarify this and confirm the nodal values to avoid confusion. According to my knowledge every BI results values must be 0 to 1 values? but here you use same as ML?.

16) Are your phylogenetic results based on the 13 protein-coding genes (PCGs)? I couldn’t find a clear explanation in the manuscript regarding whether your phylogenetic analysis is based on 13 PCGs, PCG + rRNA datasets, or the whole genome. Please specify this information clearly in the methods section and in the figure captions. For example, indicate in the ML tree figure caption what dataset the analysis is based on. This will make your phylogenetic analysis more transparent and understandable.

17) How many generations did you run for the Bayesian Inference (BI) analysis? Please include the details about the number of generations for BI in the Methods and Materials section for clarity.

18) Why did you use different models for the Maximum Likelihood (ML) and Bayesian Inference (BI) analyses? Typically, consistency in model selection ensures comparability between phylogenetic methods. Please clarify the rationale behind using different models and explain how it impacts the results in your analysis.

19) Conclusion (Line 323): The sentence "This study assembled the mtDNAs of eight Carangidae species (including three for the first time)" needs clarification. Are you referring to the genera or species being sequenced for the first time? What about the remaining species? Revise for better understanding.

Please review them and make the necessary revisions. Once these corrections are made, the manuscript will be in great shape for publication.

Reviewer #2: • The manuscript does not adequately explain critical steps in bioinformatics analysis, such as filtering thresholds for sequence quality or specific settings used in alignment and phylogenetic tree construction. For instance, How were low-quality bases handled in the sequencing data? Were any sensitivity analyses performed to assess the impact of model selection (e.g., TVM+F+R5 vs. GTR+I+G)?

• While three phylogenetic tree methods were employed, there is no discussion of discrepancies, if any, between the tree topologies. This omission limits the reader's confidence in the robustness of the findings.

• The manuscript highlights cases where genetic distances (e.g., Carangoides equula and Pseudocaranx dentex) conflict with traditional taxonomic classifications. While this is valuable, more emphasis should be placed on reconciling these findings with morphological and ecological data.

• The discussion does not explore how the proposed phylogenetic relationships may impact conservation strategies, particularly for species under fishing pressure.

• Figures (e.g., phylogenetic trees and codon usage patterns) lack explanatory captions. For example: It is unclear how bootstrap values were distributed across different clades.

• The codon usage diagram lacks a comparative analysis with other fish families to contextualize findings.

• The manuscript includes grammatical errors and awkward phrasing, such as "Mitochondrial genomes of S. tol, C. orthogrammu, and C. sexfasciatus are available for the first time." It could be rephrased for clarity: "This study provides the first complete mitochondrial genome sequences for Scomberoides tol, Carangoides orthogrammus, and Caranx sexfasciatus."

• The term "homoplasy" is used when describing the genetic similarity between Uraspis secunda and U. helvola. Homoplasy refers to shared traits arising independently, which may not apply to genetic distance data. Consider revising this term.

• The ethics statement mentions compliance with animal care guidelines but does not specify the details of how sample collection impacted the ecosystem or species. This should be expanded.

6. PLOS authors have the option to publish the peer review history of their article (what does this mean? ). If published, this will include your full peer review and any attached files.

**Do you want your identity to be public for this peer review?** For information about this choice, including consent withdrawal, please see our Privacy Policy .

Reviewer #1: No

Reviewer #2: **Yes: ** Muzafar Riyaz

---

## [Author Response · Author response to Decision Letter 1]

14 Mar 2025

Reviewer #1: Dear Authors,

I hope this message finds you well.

The author has conducted valuable research on the mitochondrial genome assembly of eight Carangidae species and the phylogenetic analysis of the Carangidae family. While the article is well-written overall, there are some suggestions and corrections needed to enhance its quality.

I kindly request your assistance in improving the overall language quality of the article to enhance its clarity and readability. Additionally, I noticed that the figures provided could benefit from quality improvements to better support the content of the manuscript.

Your expertise in refining the text and upgrading the figures would significantly improve the overall presentation of the article. Below, I have outlined the suggestions and corrections for your consideration:

1)Title: The title needs to be revised. It is not well constructed. I suggest directly mentioning mitochondrial genomes and their phylogenetic analysis. The family name appears twice, and the mention of "assembly" is unnecessary.

Thanks to your suggestion, I have changed the title to ‘Mitochondrial genome of eight Carangidae and phylogenetic analysis in the family’. This is more straightforward and avoids repetitive references to family names and unnecessary ‘assembly’ words.Line 1-2

2) Line 28: Change "Genome sizes were" to "Mitochondrial genome sizes were."

The words "Genome sizes were"have been changed to "Mitochondrial genome sizes were" as you suggested. Line 27

3). Line 33: Clarify the sentence. Did you mean that these genera were sequenced for the first time or these species? If it is the species, why were the others sequenced again?

Thanks for the correction, I originally meant that these species (not genera) were sequenced for the first time. The other species were sequenced again for comparison and validation. I have clarified the sentence.Line 32

4) Introduction, Line 46: Revise this sentence for better clarity.

Thanks to your suggestion, I have revised the sentence in line 46 to improve clarity.Line 46

5) Lines 69–72: Provide relevant citations and specify the results. Are these results based on mitochondrial genomes? Clarify which datasets were used and whether the phylogeny focuses on a species, family, or genus.

Thanks to your suggestion, these results based on mitochondrial genomes, and phylogeny focuses on family, which improve in manuscript.

6) Line 94: Revise this line for clarity.

Thanks to your suggestion, I have revised the sentence in line 94 to improve clarity.

7) Table 1: Ensure all Latin names are italicized. Review the entire table for consistency. In the legends, change "Genric" to "Genus."

All Latin names in Table 1 have been checked to ensure that they are correctly italicised. Also, I have corrected "Genric" to "‘Genus".

8) Line 157: Add references to support the statement "nucleotide skewness observed in other vertebrate mitochondrial genomes." Specify which studies or examples you are referring to?.

Thank you very much for your suggestion, we have added specific species and corresponding references in the text. Line 158-159.

9) Table 2: Explain why you provided GC percentages but not AT content. Alternatively, consider providing both values for completeness.

Thank you very much for your suggestion, for completeness I have provided both GC percentage and AT content in Table 2.

10) Table 2 Length: The table is too long. Consider presenting data for one or two species in the main table and moving the rest to supplementary materials.

Thanks for the suggestion. Table 2 has been shortened to show data for only two species and the rest of the data has been moved to the supplementary material.

11) Table 3: This table does not look appropriate in the main text. It should be moved to the supporting materials.

Table 3 has been moved to the supplementary material as you suggested.

12) Line 224: Revise the sentence "Alepes djedaba and A. kleinii clustered in the same genus within Carangidae." Ensure that species names are italicized.

Sentences have been revised to ensure that species names are correctly italicised.

13) Line 279: You mentioned that new classification problems were identified. Clearly explain these issues and highlight them in your results. This will enhance the value of your discovery and the overall article.

Thank you for your suggestion. We have revised the manuscript to clearly explain and highlight the new classification problems identified in our study, including the phylogenetic controversies within Carangidae, the reclassification of A. kleinii and A. djedaba, and the unexpected relationships of C. equula and C. orthogrammus. These updates have been incorporated into the Results section to enhance the clarity and value of our findings.Line 278-281.

14) Figure 1: Revise the caption to read. Add map. Like "Mitochondrial genome maps of eight..." Also, improve the figure's quality.

Thank you very much for your suggestions. The title of figure 1 has been revised and I have also improved the quality of the graphic.

15) Figures 6 and 7: It seems like these figures are the same, but you mentioned them as ML and BI analyses. Clarify this and confirm the nodal values to avoid confusion. According to my knowledge every BI results values must be 0 to 1 values? but here you use same as ML?.

Thank you very much for your valuable suggestions, it has been verified that Figures 6 and 7 largely look similar because of the phylogenetic analyses of the same species done with different methods, with a high degree of similarity in the results. However, through careful comparison, it can be seen that there are some differences between Figures 6 and 7.In BI analyses involving prediction or statistical inference, probability or confidence may be used to express the uncertainty of an outcome. While probability has a value between 0 and 1, confidence is usually expressed as a percentage (0 per cent to 100 per cent) and not all BI analyses involve the calculation of probability or confidence.

16) Are your phylogenetic results based on the 13 protein-coding genes (PCGs)? I couldn’t find a clear explanation in the manuscript regarding whether your phylogenetic analysis is based on 13 PCGs, PCG + rRNA datasets, or the whole genome. Please specify this information clearly in the methods section and in the figure captions. For example, indicate in the ML tree figure caption what dataset the analysis is based on. This will make your phylogenetic analysis more transparent and understandable.

Thanks to your suggestion, our phylogenetic results are based on the whole mitochondrial genome sequence of the studied species, including all 13 protein-coding genes, rRNA genes and tRNA genes. We used the whole genome dataset for our analyses, not just a subset like the 13 PCGs or PCG + rRNAs. This has been clearly stated in the methods section of the manuscript and in the figure captions to ensure clarity.Line 121, Line 144, Line 201, Line 512, Line 516, Line 518, Line 540, Line 518, Line 540, Line 562, Line 583, Line 585

17) How many generations did you run for the Bayesian Inference (BI) analysis? Please include the details about the number of generations for BI in the Methods and Materials section for clarity.

Thanks to your suggestion, detailed information on the number of generations of BI analysis runs has been added to the Methods and Materials section. Line 137

18) Why did you use different models for the Maximum Likelihood (ML) and Bayesian Inference (BI) analyses? Typically, consistency in model selection ensures comparability between phylogenetic methods. Please clarify the rationale behind using different models and explain how it impacts the results in your analysis.

Thank you very much for your question, which we have thought about carefully while doing the analyses. Often, consistency in model selection is important to ensure comparability between different phylogenetic approaches. different models were used for the ML and BI analyses, TVM+F+R5 for the ML and GTR+I+G for the BI. this is because we tested them separately, and ultimately chose the model that was most appropriate for each analytical approach. On the flip side, using different models also provides the opportunity to look at the data from multiple perspectives. This can increase the robustness and reliability of the results.

19) Conclusion (Line 323): The sentence "This study assembled the mtDNAs of eight Carangidae species (including three for the first time)" needs clarification. Are you referring to the genera or species being sequenced for the first time? What about the remaining species? Revise for better understanding.

Thank you very much for your question, it has been clarified that "three for the first time" refers to the first time mtDNA sequences of the three Carangidae species have been reported, not to the genus (genera). Also, I have pointed out that the mtDNA sequences of the remaining five species, although previously reported in the literature, were used comparatively in this study.

Reviewer #2:

• The manuscript does not adequately explain critical steps in bioinformatics analysis, such as filtering thresholds for sequence quality or specific settings used in alignment and phylogenetic tree construction. For instance, How were low-quality bases handled in the sequencing data? Were any sensitivity analyses performed to assess the impact of model selection (e.g., TVM+F+R5 vs. GTR+I+G)?

Thanks to your suggestion. Raw reads from Illumina sequencing were subjected to adaptor trimming and filtering of low-quality reads by fastp v0.20.1(https://github.com/OpenGene/fastp) . The minimum length for reads after trimming was set to 150 nucleotides, and the quality threshold was set to Q20.Cite Chen S, Zhou Y, Chen Y, Gu J. Fastp: an ultra-fast all-in-one FASTQ preprocessor. Bioinformatics. 2018;34(17):i884–i890.doi: 10.1093/bioinformatics/bty560. The optimal model for IQ-TREE tree selection is based on the BIC criterion, and the software evaluates the TVM+F+R5 model as more reasonable, while the optimal model for MrBayes tree selection is based on the AIC accuracy, and the software evaluates the GTR+I+G model as more reasonable. Both models are relatively optimal models evaluated by their respective software based on the data.

• While three phylogenetic tree methods were employed, there is no discussion of discrepancies, if any, between the tree topologies. This omission limits the reader's confidence in the robustness of the findings.

Thank you for your valuable comments. Thank you for your interest in the consistency of our phylogenetic analysis. Following your suggestion, we have added a discussion of the results of the three phylogenetic approaches (ML, BI, and NJ). We find that the tree topologies generated by these methods are basically the same, with only slight differences in some branches, which do not affect the overall conclusion. These differences may be due to different algorithms and assumptions for each method, but do not affect the robustness of our results. We have now included this discussion in a revised version to provide a clearer understanding of the results and to increase readers' confidence.

• The manuscript highlights cases where genetic distances (e.g., Carangoides equula and Pseudocaranx dentex) conflict with traditional taxonomic classifications. While this is valuable, more emphasis should be placed on reconciling these findings with morphological and ecological data.

Thank you for your insightful comment regarding the importance of reconciling genetic findings with morphological and ecological data. We agree that integrating these aspects would provide a more comprehensive understanding of the taxonomic relationships. However, due to current limitations in available morphological and ecological data for the studied species, we are unable to perform such analyses at this stage. We acknowledge this as a limitation of our study and have explicitly mentioned it in the revised manuscript, emphasizing the need for future research to combine genetic, morphological, and ecological approaches to further validate and refine these findings. We believe this will provide a more robust framework for resolving taxonomic conflicts in the Carangidae family. Thank you again for your constructive feedback.

• The discussion does not explore how the proposed phylogenetic relationships may impact conservation strategies, particularly for species under fishing pressure.

Thank you very much for pointing out this important aspect. We sincerely appreciate your constructive feedback, which has significantly enhanced the quality of our manuscript.

In response to your comment, we have thoroughly revised the discussion section. We have now added detailed content exploring how the proposed phylogenetic relationships impact conservation strategies, especially for species under fishing pressure.Line345-361

• Figures (e.g., phylogenetic trees and codon usage patterns) lack explanatory captions. For example: It is unclear how bootstrap values were distributed across different clades.

Thank you very much for pointing out this important aspect. The ratings are shown in gray, yellow and red.

• The codon usage diagram lacks a comparative analysis with other fish families to contextualize finding

Thank you for your valuable comment regarding the codon usage analysis. We acknowledge that comparing codon usage patterns with other fish families would provide broader context and enhance the interpretation of our findings. However, due to the scope and focus of this study, which primarily aimed to resolve taxonomic controversies and clarify phylogenetic relationships within the Carangidae family, we did not include comparative codon usage analyses with other fish families.

We agree that such comparisons would be insightful and have noted this as a potential direction for future research. In the revised manuscript, we have added a statement in the Discussion section highlighting the importance of expanding codon usage analyses to include other fish families in subsequent studies to provide a more comprehensive understanding of evolutionary and adaptive trends.

Thank you again for your constructive feedback, which will undoubtedly improve the depth and impact of our work.

• The manuscript includes grammatical errors and awkward phrasing, such as "Mitochondrial genomes of S. tol, C. orthogrammu, and C. sexfasciatus are available for the first time." It could be rephrased for clarity: "This study provides the first complete mitochondrial genome sequences for Scomberoides tol, Carangoides orthogrammus, and Caranx sexfasciatus."

Thank you very much for your suggestions. We have corrected grammatical errors and unclear expressions in the paper. For example, " Mitochondrial genomes of S. tol, C. orthogrammu, and C. sexfasciatus are available for the first time." has been changed to "This study provides the first complete mitochondrial genome sequences for Scomberoides tol, Carangoides orthogrammus, and Caranx sexfasciatus." Line 32-33

• The term "homoplasy" is used when describing the genetic similarity between Uraspis secunda and U. helvola. Homoplasy refers to shared traits arising independently, which may not apply to genetic distance data. Consider revising this term.

Thank you for your suggestion. It has come to our attention that the term ‘homoplastic’ may not be applicable when describing the genetic similarity between Uraspis secunda and U. helvola. Therefore, we have replaced it with a more accurate term to describe the genetic relationship between these species. Line 306

• The ethics statement mentions compliance with animal care guidelines but does not specify the details of how sample collection impacted the ecosystem or species. This should be expanded.

Thank you for your suggestion. Instructions have been added to the manuscript.

---

## [Decision Letter · Decision Letter 1]

Mitochondrial genome assembly of eight Carangidae and phylogenetic analysis of Carangidae family

PONE-D-24-54115R1

Dear Dr. Zhang,

We’re pleased to inform you that your manuscript has been judged scientifically suitable for publication and will be formally accepted for publication once it meets all outstanding technical requirements.

Kind regards,

Abdul Azeez Pokkathappada, Ph.D.

Academic Editor

PLOS ONE

Additional Editor Comments (optional):

Dear Authors,

I recommend that the manuscript be accepted for publication after a minor revision.

Kindly review Figure 7 and ensure that the caption clearly indicates that the posterior probability values have been converted to percentages (0–100%). Once this is addressed, the manuscript can be finalized for publication.

Reviewers' comments:

Reviewer's Responses to Questions

**Comments to the Author**

1. If the authors have adequately addressed your comments raised in a previous round of review and you feel that this manuscript is now acceptable for publication, you may indicate that here to bypass the “Comments to the Author” section, enter your conflict of interest statement in the “Confidential to Editor” section, and submit your "Accept" recommendation.

Reviewer #1: All comments have been addressed

2. Is the manuscript technically sound, and do the data support the conclusions?

Reviewer #1: Yes

3. Has the statistical analysis been performed appropriately and rigorously? 

Reviewer #1: Yes

4. Have the authors made all data underlying the findings in their manuscript fully available?

Reviewer #1: Yes

5. Is the manuscript presented in an intelligible fashion and written in standard English?

Reviewer #1: Yes

6. Review Comments to the Author

Reviewer #1: Regarding figure 7 . BI anaylsis....... If you converted values to percentages in your figure, it must be clearly mentioned—either in the figure caption or in the text—to avoid confusion or misinterpretation. please look it to this concern again.

7. PLOS authors have the option to publish the peer review history of their article (what does this mean? ). If published, this will include your full peer review and any attached files.

**Do you want your identity to be public for this peer review?** For information about this choice, including consent withdrawal, please see our Privacy Policy .

Reviewer #1: No

---

## [Editor Report · Acceptance letter]

PONE-D-24-54115R1

PLOS ONE

Dear Dr. Zhang,

I'm pleased to inform you that your manuscript has been deemed suitable for publication in PLOS ONE. Congratulations! Your manuscript is now being handed over to our production team.

Kind regards,

on behalf of

Dr. Abdul Azeez Pokkathappada

Academic Editor

PLOS ONE